# Identification and Validation of Th1-Selective Epitopes Derived from Proteins Overexpressed in Breast Cancer Stem Cells

**DOI:** 10.3390/vaccines13050525

**Published:** 2025-05-15

**Authors:** Denise L. Cecil, Daniel Herendeen, Meredith Slota, Megan M. O’Meara, Yushe Dang, Lauren Corulli, Mary L. Disis

**Affiliations:** Cancer Vaccine Institute, University of Washington, Seattle, WA 98109, USAydang@uw.edu (Y.D.); lrr2@uw.edu (L.C.);

**Keywords:** T-helper 1, vaccine, epitopes, cancer stem cells, epithelial-to-mesenchymal transformation

## Abstract

Background: Breast cancer stem cells (CSCs), particularly those enriched in triple-negative breast cancer (TNBC), are key contributors to tumor recurrence, metastasis, and resistance to therapy. CSCs often undergo epithelial-to-mesenchymal transformation (EMT), enhancing their invasiveness. Immune-based strategies that selectively target CSC/EMT antigens offer a promising therapeutic approach. Methods: Twelve candidate CSC/EMT-associated proteins were identified through a systematic literature review. Human serum samples were assessed for antigen-specific IgG using ELISA. Th1/Th2 cytokine profiles, in response to predicted MHC II epitopes, were measured by ELISPOT in PBMCs. Epitope immunogenicity and tumor inhibition were evaluated in murine models, using either TNBC or luminal B syngeneic breast cancer cell lines. Results: Six of the candidate proteins (SOX2, YB1, FOXQ1, MDM2, CDH3, CD105) elicited antigen-specific IgG in human serum. Th1-selective epitopes, defined by high Th1/Th2 ratios, were identified for five of these proteins. Immunization of mice with peptide pools derived from CD105, CDH3, MDM2, SOX2, and YB1 induced significant antigen-specific IFN-γ responses. Tumor growth was significantly inhibited in the vaccinated mice across both the TNBC and luminal B breast cancer models, with mean tumor volume reductions ranging from 61% to 70%. Conclusions: CSC/EMT-associated antigens are immunogenic in humans and can be targeted using Th1-selective epitope-based vaccines. Immunization with these epitopes effectively inhibits tumor growth in multiple murine models of breast cancer. These findings support further clinical evaluation of CSC/EMT-targeted vaccines, especially for high-risk or advanced-stage breast cancer patients.

## 1. Introduction

Standard therapies for breast cancer are not effective at eliminating all cancer cells in tumors, particularly breast cancer stem cells (CSCs). CSCs are thought to be significant drivers of tumor growth due to their extensive self-renewal, leading to drug resistance, recurrence, and metastasis [1,2]. During metastasis, breast CSCs can infiltrate the surrounding tissue or enter the circulation by epithelial-to-mesenchymal transformation (EMT), a process by which epithelial cell polarity and intercellular adhesion properties are lost, and the cells gain migratory and invasive properties [3].

Several studies have established that the CSC cell population is enriched in triple-negative breast cancer (TNBC) tissues and cell lines, using both the unique stem cell marker ALDH1 [4] and the tumor cell phenotypic marker of CD44^+^/CD24^−^ [5,6]. Additionally, people with a genetic predisposition to develop cancer, such as those with BRCA1 and BRCA2 mutations, are more likely to have “stem cell”-like tumors [7]. If a vaccine could be developed to eliminate CSC, it could be used to prevent breast cancer recurrence or even prevent breast cancer.

The presence of type I tumor-infiltrating lymphocytes has been associated with an improved prognosis in breast cancer, particularly in TNBC and HER2^+^ disease [8]. Our group and others have discovered that within the natural sequence of non-mutated tumor-associated antigens are MHCII-restricted epitopes that selectively generate either an IFN-γ or an IL-10 immune response [9,10,11]. Vaccination with a pool of only IFN-γ- or IL-10-selective epitopes or with a pool of both types of epitopes targeting IGFBP-2 in a mouse model of breast cancer or PSMA in a mouse model of prostate cancer revealed that it is essential for the IL-10 selective epitopes to be edited out of a polyepitope vaccine. Tumor growth was only inhibited in the mice receiving the Th1 (IFN-γ secreting)-selective vaccine in both models. Furthermore, if the IFN-γ- or IL-10-selective epitopes were mixed in a vaccine in either model, tumor growth was no different than in the mice receiving the sham vaccine due to the immune suppressive effects of IL-10 secretion [10,11]. These data underscore the necessity to edit out epitopes that would self-regulate or dampen the IFN-γ response.

Our goal was to develop a vaccine targeting breast cancer stem cells. By identifying CSC/EMT proteins associated with different cellular pathways, defining those proteins as antigenic, identifying Th1-selective epitopes from each protein, and assessing in vivo immunogenicity and clinical efficacy, we defined epitopes from five antigens to include in a multi-epitope, multi-antigen vaccine.

## 2. Materials and Methods

### 2.1. Candidate Antigen Identification

We performed a systematic literature review in Pubmed Central using the following key words in various combinations: breast cancer, stem cell, EMT, embryogenesis, protein expression, biomarker, multivariate, prognosis, outcome, relapse, survival, and metastasis to identify proteins associated with cancer stem cells. Ultimately, 90 potential protein targets were identified. The proteins were evaluated further with an additional literature search, using the protein name plus the keywords “breast cancer” and “prognosis” or “prognostic”. A total of 1158 articles were evaluated for relevance and 321 relevant journal articles were identified and reviewed in detail. Proteins were prioritized for further study based on the following characteristics: (1) upregulated or overexpressed in breast cancer, (2) associated with EMT, (3) associated with breast cancer stem cells, and (4) shown to be an independent poor-prognostic indicator in univariate and/or multivariate analysis for some solid tumor. Twelve candidate proteins were selected for evaluation: CD105, CDC25B, CDH3, FOXQ1, ID1, MDM2, PRL3, SATB1, SIX1, SNAIL, SOX2, and YB1 (Figure 1).

### 2.2. Human Subjects

The studies were approved by the University of Washington Human Subjects Division (protocol code STUDY00001570 and date of approval, 13 December 2024). All samples were collected after informed consent. Plasma samples were obtained from 100 female volunteer blood donors at the Puget Sound Blood Center in Seattle, WA (median age: 51 years; age range: 33–73) and from 153 female breast cancer patients, including 82 Estrogen Receptor (ER)+, 50 HER2-, and 21 ER-HER2- cases. These patients, who had consented to participate in the Fred Hutchinson Cancer Center/University of the Washington Breast Specimen Repository and Registry (IR file #5306), included 121 with stage I/II and 32 with stage III/IV disease (median age: 52 years; age range: 33–89). Peripheral blood mononuclear cells (PBMCs) were collected and cryopreserved from 20 female control volunteers (median age: 49; age range: 18–79) and 21 female breast cancer patients (median age: 44; age range: 19–79) [12].

### 2.3. Analysis of Antibody Immunity

IgG antibodies specific to the candidate target proteins were measured using an indirect ELISA, as previously described, with modifications [13]. Odd columns (1–9) of microtiter plates were coated with carbonate buffer alone and even columns (2–10) with carbonate buffer containing 0.1 µg/mL human recombinant protein. Plates were washed and blocked with PBS/5% BSA for one hour at room temperature. Patient sera (diluted 1:200 in PBS/1% BSA) was added and incubated for one hour at room temperature. Plates were washed, incubated with anti-human IgG-HRP conjugate, and developed with TMB reagent. Optical density (OD) values were background-corrected by subtracting the OD of the buffer-coated wells from that of the protein-coated wells, and the data were reported as µg/mL of antigen-specific IgG. The incidence was calculated from every non-zero number.

### 2.4. Analysis of Th1-Selective Epitope-Specific T-Cell Responses

Multiple publicly-available web-based algorithms were used to predict epitopes from each candidate target protein that would bind human MHC II with high affinity across several HLA-DR alleles (Appendix A) [14]. The peptides were synthesized and determined to be >90% pure by HPLC (CPC Scientific, Milpitas, CA). An antigen-specific IFN-gamma (γ; Mabtech Cat# 3420-3-1000, RRID:AB_907282 and Cat# 3420-6-1000, RRID:AB_907272) or IL-10 (Mabtech Cat# 3430-3-1000, RRID:AB_907306 and Cat# 3430-6-1000, RRID:AB_2125387) response was evaluated for each epitope in human PBMC via ELISPOT. Murine splenocytes were assessed by ELISPOT for antigen-specific IFN-γ (Mabtech Cat# 3321-3-1000, RRID:AB_907278 and Cat# 3321-6-1000, RRID:AB_907270) [11]. Statistically significant differences (*p* < 0.05) between the mean spot count in the experimental wells and the mean spot count in the no antigen control wells defined a positive response. The results are presented as the ratio of the product of percent incidence and mean magnitude. Th1-selective epitopes were defined by ratios > 4 with fewer than 10% of the donors generating an IL-10 response, while Th2-selective epitopes had ratios < 1. Epitopes with intermediate values were classified as eliciting a mixed Th1/Th2 response [10].

Peptide specific T-cells were generated via short term T-cell culture and then documented to respond to corresponding protein by ELISPOT using our previously published methods [15]. Restriction and specificity of the antigen-specific T-cells to Class II were validated by blocking either the MHC I or MHC II molecules with 1 µg/mL of Ultra-LEAF purified anti-human HLA-A,B,C (BioLegend Cat# 311441, RRID:AB_2800814) or Ultra-LEAF purified anti-human HLA-DR (BioLegend Cat# 307665, RRID:AB_2800796) and stimulating with appropriate antigen assaying for the presence or absence of antigen-specific cytokine release by the generated T-cell lines.

### 2.5. Animal Models and Syngeneic Tumor Cell Lines

Animal care and use were in accordance with University of Washington IACUC guidelines (Protocol # 2878-01 and date of approval 3/27/25). Female, FVB/N, FVB/N-TgN (MMTVneu)-202Mul mice (TgMMTV-neu) (Jackson Laboratory) or FVB-Tg (C3-1-TAg)cJeg/Jeg (C3(1)-Tag) mice (provided by Dr. Jeff Green, NCI) were used in this study. All mice were 6–8 weeks old at the study’s start. A power analysis established that five mice per group would provide 80% power to detect a significant pairwise difference in tumor volume at the two-sided alpha level of 0.05. Neither the investigators nor animal technicians were blinded to the treatment groups. Before initiating the experiment, the mice were randomly assigned to different treatment groups, and procedures were performed in random order. The studies were terminated when the mean tumor volume in the vaccinated and control groups were significantly different for at least two measurements.

The mouse mammary tumor cell lines, MMC, derived from TgMMTV-neu spontaneous mammary tumors [16], and M6, derived from C3(1)-Tag spontaneous mammary tumors [17], were authenticated before in vivo implant. MMC was confirmed to express rat neu, and M6 was verified to express the SV40 antigen and low levels of the estrogen receptor (Appendix A).

### 2.6. Vaccination and Assessment of Tumor Growth

Mice were immunized subcutaneously using a 26 ^1^/_2_ G needle. Each mouse was injected with a pool of peptides (50 μg each) derived from either CD105 (p569/p603/p626), CDH3 (p93/p137), MDM2 (p57/p80/p97/p104), SOX2 (p193/p217), or YB1 (p138) in complete Freund’s adjuvant/incomplete Freund’s adjuvant (Sigma-Aldrich, St. Louis, MO), and adjuvant alone was used as a control group. Four immunizations were given two weeks apart [10]. Two weeks after the last vaccine, splenocytes were collected for immunogenicity studies, or the syngeneic mouse cell line appropriate to that model (0.5 × 10^6^ cells) was implanted into the flank of the animal ) [11]. Tumors were measured as previously described [14]. All tumor growth is presented as mean tumor volume (mm^3^ ± SEM).

### 2.7. Statistical Analysis

Model assumptions were checked using the Shapiro–Wilk normality test and by visual inspection of the residual and fitted value plots. The unpaired, two-tailed Student’s *t*-test and one-way ANOVA with Tukey’s post-hoc test were used to evaluate differences when normality was confirmed. When the normality of the data was not confirmed, the non-parametric Kruskal–Wallis and Mann–Whitney tests were used. Differences in tumor volume were determined by two-way ANOVA with a Dunnett post-test for multiple comparisons. A *p* value of <0.05 was considered significant (GraphPad Software, Prism v.10).

## 3. Results

### 3.1. Antibodies Specific to CSC/EMT Candidate Target Proteins Can Be Detected in the Sera of Both Volunteer Donors and Breast Cancer Patients

To determine whether the candidate target proteins were human immunogens, we assayed human sera from both breast cancer and volunteer donors for humoral immune responses. Because there was no statistical difference in either the incidence of response or IgG levels between the breast cancer and volunteer donors, we did not separate them, and we present the pooled values (Figure 2). We were able to detect measurable quantities of antigen-specific IgG to six of the 12 candidate target proteins. We observed antigen-specific IgG in 95% of donors to SOX2 (mean, 0.92 ± 0.06 µg/mL; Figure 2A), 70% of donors to YB1 (mean, 0.78 ± 0.12 µg/mL, Figure 2B), 75% of donors to FOXQ1 (mean, 0.243 ± 0.02 µg/mL, Figure 2C), 38% of donors to MDM2 (mean, 0.189 ± 0.03 µg/mL, Figure 2D), 53% of donors to CDH3 (mean, 0.15 ± 0.02 µg/mL, Figure 2E), and 28% of donors to CD105 (mean, 0.15 ± 0.02 ng/mL, Figure 2F). Only those proteins shown to be human antigens were prioritized for further evaluation (Figure 1).

### 3.2. Th1-Selective Epitopes Can Be Identified from CSC/EMT Antigens and Are Class II Restricted

T-cells that secreted IFN-γ or IL-10 in response to specific epitope stimulation of human PBMC were identified [10,11]. We have previously demonstrated the most clinically effective epitopes to include in a vaccine elicited high levels of IFN-γ secretion in the greatest number of people, with little to no IL-10 secretion [10,11]. These epitopes, which would be defined by high Th ratios, were termed as Th1-selective. The distribution of responses was antigen-dependent. One hundred percent (6/6) of the epitopes evaluated for YB1 were Th1-selective (Th ratio range, 5.8–1980; Figure 3A). Seven of the eight epitopes assessed for MDM2 were Th1-selective (Th ratio range, 35–313) and one peptide (p30) generated a mixed response (Figure 3B). Seven of 10 epitopes located in either the C- or N-terminus of CD105 were determined to be Th1-selective, whereas the remining three epitopes in the middle of the protein generated a predominant Th2 response (p302 and p316) or no response (p258; Figure 3C). For the epitopes derived from CDH3, 73% (8/11) predominantly secreted IFN-γ, whereas two epitopes generated mixed responses, and one generated no response (Figure 3D). Three of the five evaluated epitopes from SOX2 selectively secreted IFN-γ, and the remaining two epitopes secreted both IFN-γ and IL-10 equally (Figure 3E). Only 33% (2/6) of the epitopes assessed for FOXQ1 were Th1-selective, whereas 67% (4/6) generated mixed responses, Th2-selective responses, or no response (Figure 3F). Because FOXQ1 contained so few Th1-selective epitopes and the signaling pathways in which this protein is associated, such as Wnt/β-catenin [18] and the MEK-ERK2 regulatory axis [19], are also associated with SOX2 [20] and YB1 [21], we chose not to consider this antigen further due to its functional redundancy with others (Figure 1).

We confirmed that the in situ-predicted MHCII-restricted epitope was a sequence that was naturally processed and presented in the native protein. The ability of peptide-specific T-cells to respond to protein was validated for a representative epitope in each antigen. The epitope-specific T-cells secreted significantly more IFN-γ to the recombinant protein from which that epitope was derived for all antigens tested, suggesting that all sequences were indeed native epitopes (Appendix A). Furthermore, the IFN-γ response was MHCII-restricted. The epitope- and/or protein-specific responses for CD105-p569 (Appendix A), CDH3-p137 (Appendix A), MDM2-p80 (Appendix A), SOX2-p217 (Appendix A), and YB1-p138 (Appendix A) were significantly inhibited in the presence of an MHCII-blocking antibody (*p* < 0.05 for all). No inhibition of the response was observed in the presence of an MHCI-blocking antibody, except for the MDM2-p80-specific response to MDM2 protein (Appendix A; *p* < 0.0001). All antigens were considered further for in vivo testing (Figure 1). The epitopes are highly homologous between mouse and man (Appendix A).

### 3.3. CSC/EMT Th1-Selective Epitope-Based Vaccines Are Immunogenic and Inhibit Tumor Growth in Murine Breast Cancer Models

FVB/N mice were vaccinated with a peptide pool of each single antigen (Figure 4A), and IFN-γ ELISPOT analysis revealed that the mice developed immune responses to all antigens. There was a significant IFN-γ response in the mice immunized with the peptide pool of CD105 (mean, 35 ± 11 corrected spots per well (cSPW)/10^6^ cells; *p* = 0.007), CDH3 (mean, 464 ± 104 cSPW/10^6^ cells; *p* = 0.005), MDM2 (mean, 220 ± 54 cSPW/10^6^ cells; *p* = 0.004), SOX2 (mean 639 ± 39 cSPW/10^6^ cells; *p* < 0.0001), and YB1 (mean 236 ± 75 cSPW/10^6^ cells; *p* = 0.019) as compared with the control (Figure 4B).

Mice immunized with Th1-selective epitopes from each single antigen (Figure 4C) demonstrated reduced breast cancer growth. For the vaccination targeting MDM2 and YB1, we used the luminal breast cancer TgMMTV-neu mouse model, and for the vaccination targeting CD105, CDH3, and SOX2, we used the TNBC C3(1)-Tag mouse model. Antigen expression was documented in the tumor cell lines by Western blot prior to cell implant. After vaccination, the tumor volume was significantly reduced by a mean of 70 ± 15% for the MDM2 peptides (*p* = 0.0002; Figure 4D), by a mean of 67 ± 15% for the YB1 peptides (*p* = 0.0007; Figure 4E), by a mean of 69 ± 4% for the CD105 peptides (*p* < 0.0001; Figure 4F), by a mean of 63 ± 20% for the CDH3 peptides (*p* < 0.0001; Figure 4G), and by a mean of 61 ± 6% for the SOX2 peptides (*p* < 0.0001; Figure 4H).

## 4. Discussion

Immunologically eliminating CSC/EMT could prevent the development of cancer, improve the efficacy of standard chemotherapies, and inhibit metastatic spread [1]. The data we present here demonstrate that CSC/EMT-associated proteins are human immunogens, Th1-selective epitopes can be derived from these proteins, and vaccination with these epitopes can inhibit the growth of mammary cancer in vivo.

We determined that six of the CSC/EMT proteins that we identified as overexpressed and poor prognostic indicators in breast cancer were human antigens. SOX2, YB1, and FOXQ1 are all transcription factors associated with signaling pathways that are constitutively activated in breast cancer, particularly TNBC, including Wnt/B-catenin, Hedgehog, and Notch, which regulate the maintenance and survival of CSCs [21,22,23,24]. The importance of the Wnt/B-catenin, Hedgehog, and Notch signaling pathways driving cancer have been appreciated recently. Several clinical trials have begun to assess the activity of small-molecule agents that antagonize these pathways [25,26,27]. MDM2, an E3 ubiquitin ligase, is also an intracellular antigen and can target the tumor suppressor protein p53 and other substrates for degradation via the proteasome [28]. Finally, two transmembrane proteins were identified as antigens: CDH3, an instrumental protein in modulating tumor cell metastasis [29] and CD105, which is expressed on the endothelial cells of tumoral blood vessels and stroma and is associated with angiogenesis [30].

Th1-selective epitopes were identified for all six candidate antigens. However, only five of the antigens contained consecutive Th1-selective epitopes with minimal untested sequences in between the epitopes, allowing for the formulation of extended multi-epitope vaccines. Longer MHCI epitopes have been shown to be more immunogenic [31]. When an epitope length was extended from eight amino acids to 30 amino acids, there was a greater than two-fold increase in cytotoxicity, presumably due to the fact that shorter peptides could directly bind to the MHC of non-professional APCs, which lack the necessary costimulatory signals required for complete T cell activation, and the fact that longer peptides have to processed and presented by professional APC in a pro-inflammatory environment like the lymph node [31]. The MHCII epitopes described in this study are promiscuous, high-affinity binding epitopes that are predicted to bind across multiple HLA-DR alleles, and the peptide length does not have to be as strict as for MHCI epitopes. However, being able to formulate a multi-epitope MHCII vaccine would allow for even more flexibility of allelic differences, highlighting the more universal utility of this vaccine.

Immunization with an IFN-γ-selective vaccine targeting CSC/EMT antigens could have significant clinical therapeutic impacts across all breast cancer subtypes. Similar to our in vivo study in the C3-Tag basal breast cancer model, other studies have targeted single CSC/EMT antigens in the basal-like breast cancer model TUBO and/or the triple-negative breast cancer model 4T1 in BALB/c mice. Targeting the CSC/EMT antigen Cripto-1 (Cr-1) in the 4T1 model or the xCT protein, also known as SLC7A11, in the 4T1 or TUBO models with a plasmid-based vaccine inhibited tumor growth by about 30–40% and significantly reduced metastatic spread to the lungs [32,33]. We also show that a vaccine targeting CSC/EMT proteins can be used to inhibit tumor growth in a luminal B breast cancer model. Luminal B breast cancer has been shown to have lower levels of CSC than other breast cancer subtypes [34], suggesting that this vaccine has broader uses than in just CSC/EMT-high tumors. Furthermore, a vaccine targeting CSC/EMT could be used as a viable therapy to prevent breast cancer in genetic high-risk individuals, such as those with BRCA1/2 mutations. Loss of BRCA function leads to genomic instability and potentiates the oncogenic conversion of tissue stem cells to cancer stem cells [35].

## 5. Conclusions

The identification and validation of Th1-selective epitopes targeting CSC/EMT in animal models has laid the groundwork for translation to humans. The vaccine caused no toxicity in the immunized mice, including no autoimmune toxicity, allowing the development of a Phase I clinical trial in patients with advanced-stage triple-negative and hormone receptor-positive breast cancer (NCT02157051).

## Figures and Tables

**Figure 1 vaccines-13-00525-f001:**
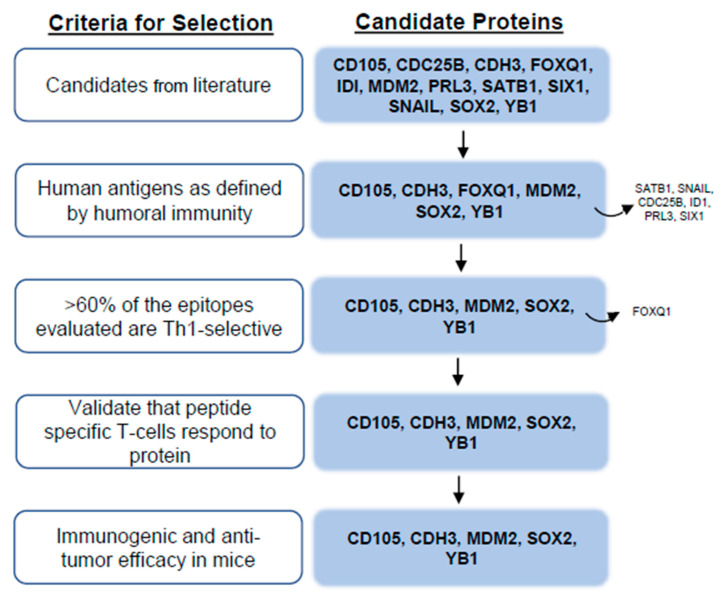
Flow chart for antigen identification.

**Figure 2 vaccines-13-00525-f002:**
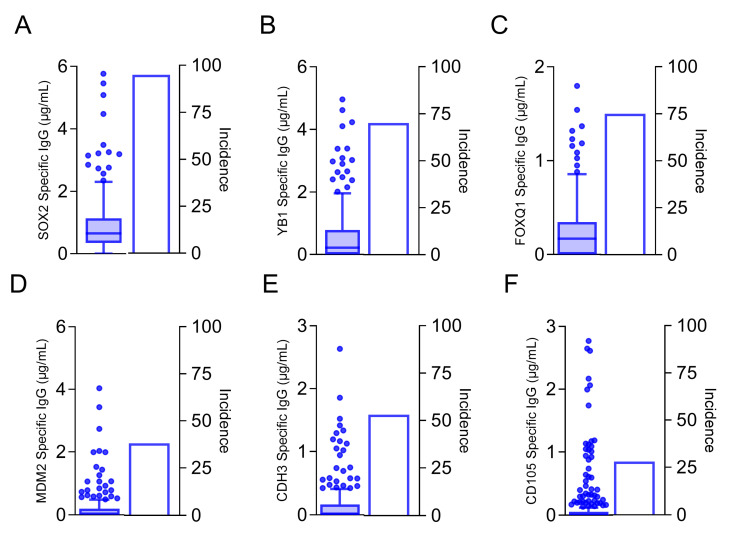
Antibodies specific to cancer stem cell/EMT antigens can be detected in the sera of both volunteer donors and breast cancer patients. Antigen-specific IgG (µg/mL; left *y*-axis) are presented as box and whisker plots with Tukey outliers and percent incidence (right *y*-axis) for (**A**) SOX2, (**B**) YB1, (**C**) FOXQ1, (**D**) MDM2, (**E**) CDH3, and (**F**) CD105. n = 253.

**Figure 3 vaccines-13-00525-f003:**
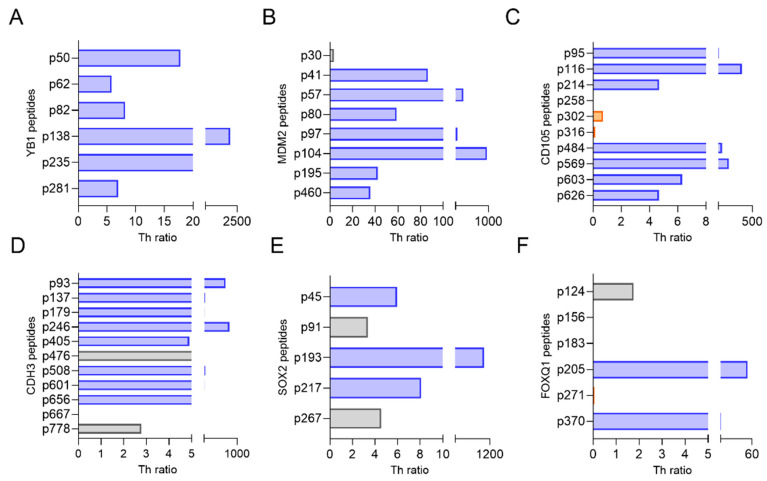
Th1-selective epitopes can be identified from CSC/EMT antigens. Th ratio (*x*-axis) calculated from PBMC stimulated with epitopes (*y*-axis) derived from (**A**) YB1, (**B**) MDM2, (**C**) CD105, (**D**) CDH3, (**E**) SOX2, and (**F**) FOXQ1. Blue bars denote Th1-selective epitopes, gray bars denote a mixed Th1 and Th2 response, and orange bars denote a Th2-selective response. n = 10 volunteer donors and 10 breast cancer donors.

**Figure 4 vaccines-13-00525-f004:**
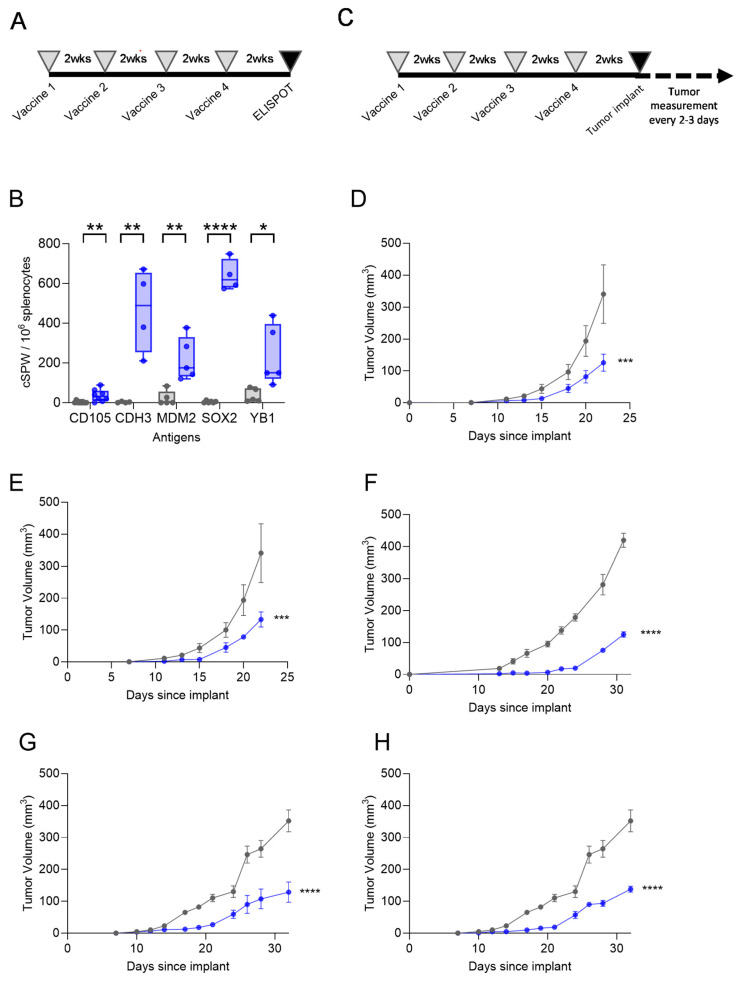
CSC/EMT Th1-selective epitope-based vaccines are immunogenic and inhibit tumor growth in murine breast cancer models. (**A**) Experimental design schematic for immunogenicity studies. (**B**) IFN-γ corrected spots per well (cSPW) per 10^6^ splenocytes after immunization with the peptide vaccine (blue bars) or control (gray bars), presented as box and whisker plots with a horizontal line at the median and whiskers at minimum and maximum, showing all points. (**C**) Experimental design schematic for tumor challenge studies. Mean tumor volume (mm^3^ ± SEM) for the mice immunized with the control (gray lines) or peptide (blue lines) vaccines derived from (**D**) MDM2, (**E**) YB1, (**F**) CD105, (**G**) CDH3, and (**H**) SOX2. The control curves are the same for panels (**D**,**E**) and the same for panels (**G**,**H**). * *p* < 0.05, ** *p* < 0.01, *** *p* < 0.001, **** *p* < 0.0001.

## Data Availability

Data are available upon reasonable request.

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
