# Peer review of "Identification and Validation of Th1-Selective Epitopes Derived from Proteins Overexpressed in Breast Cancer Stem Cells"

_vaccines, 2025, doi:10.3390/vaccines13050525_

Round 1
Reviewer 1 Report
Comments and Suggestions for Authors
This study explored a potential vaccine strategy to combat breast cancer by targeting cancer stem cells (CSCs), which are responsible for tumor growth, recurrence and are typically resistant to traditional treatment. The author has identified 12 proteins lined to CSC/EMT, six of which are antigen. The Th1-selective, multi-epitope vaccines were further validated using five of these antigens, and preclinical models demonstrated that the vaccine effectively activated the immune system and significantly inhibited tumor growth. Before acceptance, several issues should be addressed as follows:
- The resolution and clarity of all Figures in the manuscript should be improved with much clear readability.
- In ‘Animal models and syngeneic tumor cell line’ section, the author mentioned that ‘The MMC cell line was verified to express rat neu by flow cytometry and the M6 cell line was verified to express the SV40 antigen by Western blot and low levels the estrogen receptor by RT-PCR’. It is better to provide the original data, such as representative flow cytometry histograms, Western blot images, and RT-PCR amplification curves or gel electrophoresis result, to strengthen the manuscript.
- In Figure 4, it would be helpful to include a schematic outline of in vivo mice experiment for understanding; How about the body weight change of mice throughout the experiment?
Author Response
Comment 1: The resolution and clarity of all Figures in the manuscript should be improved with much clear readability.
Response 1: The figures in the revised manuscript are of higher resolution with much clearer readability.
Comment 2: In ‘Animal models and syngeneic tumor cell line’ section, the author mentioned that ‘The MMC cell line was verified to express rat neu by flow cytometry and the M6 cell line was verified to express the SV40 antigen by Western blot and low levels the estrogen receptor by RT-PCR’. It is better to provide the original data, such as representative flow cytometry histograms, Western blot images, and RT-PCR amplification curves or gel electrophoresis result, to strengthen the manuscript.
Response 2: A new Figure S1 has been added into the supplementary figures. This figure includes the validation data as suggested and referred to in the main text in lines 150-152.
Comment 3: In Figure 4, it would be helpful to include a schematic outline of in vivo mice experiment for understanding; How about the body weight change of mice throughout the experiment?
Response 3: Schematics for both immunogenicity and tumor challenges studies are now included in Fig. 4 and referred in the text on lines 306 and 314.
Reviewer 2 Report
Comments and Suggestions for Authors
In this original research article, the authors explore the option of eliciting tumor growth inhibition by designing a vaccine that elicits IFNgamma secretion and hence directs the immune response towards the development of a robust Th1-mediated response, to immunologically eliminate breast cancer stem cells (or ultimately other breast cancer cells) via vaccination. In their approach, they first perform a systematic literature review and identify 12 candidate target proteins that are overexpressed in breast cancer, and select 6 of those as antigens defined by humoral immunity as confirmed by the presence of antigen specific IgG in human sera. They select the peptides predicted to promiscuously bind human MHC II using bioinformatic tools, generate peptide specific T-cells via short term T-cell culture and then determine their response to the corresponding protein by ELISPOT. So they identify the Th1-selective epitopes. Using antibody-mediated MHCI or II inhibition, they demonstrate that the response is MHCII restricted, as the epitope- and protein-specific responses could mostly be blocked with an anti-MHCII antibody but was mostly not affected with MHC I blocking antibody, except in one example.
The potential of these peptides for immunization was tested in an in vivo mouse experiment. For vaccination addressing MDM2 and YB1, the luminal breast cancer TgMMTV-neu mouse model was used, and to address CD105, CDH3 and SOX2, the triple negative breast cancer C3(1)-Tag mouse model was used. Immunization with Th1-selective epitopes inhibited breast cancer growth. A significant IFN-gamma response was achieved in all immunized groups compared with the controls. After vaccination, the tumor volumes were significantly reduced in comparison with the control groups.
This is a well-designed study with important outcomes with a high translational value, as the immunization with an IFN-γ-selective vaccine could have significant clinical therapeutic impact across several breast cancer subtypes, and even be used prophylactically in high-risk individuals. This is also confirmed in the concluding paragraph which states that the identification and validation of Th1-selective epitopes has paved the way to a Phase I clinical trial, which then uses plasmid-based vaccines.
Please find below a list of remarks which I hope you will find helpful.
Lines 108-110: all commercially acquired antibodies should be cited with RRIDs.
Line 133: “Four immunizations were given two weeks apart “ - regimen of the vaccination should be put into the temporal context of tumor cell line introduction, please add this information.
Line 136, mm3, 3 in superscript
Line 144: differences in tumor volumes were determined
Line 177: “a predominant Th2 response (p302 and p316)” – should there not ne an orange bar near p316 in 3C?
Line 204: In the text the MHCI blocking is stated to be inhibitory for MDM2-p80, but according to the Figure S1C the MDM2-protein activity is inhibited here. MDM2-p80 does not look so strongly inhibited by the anti-MHCII antibody, please explain.
Line 231:” (D) CD105 (D) CDH3 and (E) SOX2”: should be (D), (E) and (F)
Figure 4: data presentation: are the control curves the same data in B and C / E and F? This should be mentioned in the Figure legend.
Author Response
Comment 1: Lines 108-110: all commercially acquired antibodies should be cited with RRIDs.
Response 1: RRIDs have been included for all described antibodies (see lines 116-120 and lines 132-134).
Comment 2: Line 133: “Four immunizations were given two weeks apart “ - regimen of the vaccination should be put into the temporal context of tumor cell line introduction, please add this information.
Response 2: Tumor implant is now described in line 158-161. Additionally, schematics for both immunogenicity and tumor challenges studies are now included in Fig. 4 and referred in the text on lines 306 and 314.
Comment 3: Line 136, mm3, 3 in superscript
Response 3: Changed to superscript, line 162
Comment 4: Line 144: differences in tumor volumes were determined
Response 4: Revised, lines 167-168.
Comment 5: Line 177: “a predominant Th2 response (p302 and p316)” – should there not ne an orange bar near p316 in 3C?
Response 5: Yes, there should be an orange bar for this epitope. Unfortunately, the resolution of the figures in the original submission was low and the small orange bar did not show up. The new figures are of higher resolution and the orange bar is now visible.
Comment 6: Line 204: In the text the MHCI blocking is stated to be inhibitory for MDM2-p80, but according to the Figure S1C the MDM2-protein activity is inhibited here. MDM2-p80 does not look so strongly inhibited by the anti-MHCII antibody, please explain.
Response 6: We described the response as “epitope- and/or protein-specific responses” for each antigen-specific T-cell line. We added a clearer description of the exception for MDM2-p80, lines 262-264.
Comment 7: Line 231:” (D) CD105 (D) CDH3 and (E) SOX2”: should be (D), (E) and (F)
Response 7: Figure 4 now has 2 added panels, the labeling has been revised, line 360.
Comment 8: Figure 4: data presentation: are the control curves the same data in B and C / E and F? This should be mentioned in the Figure legend.
Response 8: Yes, the controls are common for some of the antigens. This has been mentioned in the legend, line 360.